# Recurrent Flow Update Model Using Image Pyramid Structure for 4K Video Frame Interpolation

**DOI:** 10.3390/s25010290

**Published:** 2025-01-06

**Authors:** Sangjin Lee, Chajin Shin, Hong-Goo Kang, Sangyoun Lee

**Affiliations:** School of Electrical and Electronic Engineering, Yonsei University, Seoul 03722, Republic of Korea; sglee97@yonsei.ac.kr (S.L.); chajin@yonsei.ac.kr (C.S.); hgkang@yonsei.ac.kr (H.-G.K.)

**Keywords:** video frame interpolation, end-to-end learning, hierarchical flow refinement, difference map

## Abstract

Video frame interpolation (VFI) is a task that generates intermediate frames from two consecutive frames. Previous studies have employed two main approaches to extract the necessary information from both frames: pixel-level synthesis and flow-based methods. However, when synthesizing high-resolution videos using VFI, each approach has its limitations. Pixel-level synthesis based on the transformer architecture requires high complexity to achieve 4K video results. In the case of flow-based methods, forward warping can produce holes where pixels are not allocated, while backward warping approaches struggle to obtain accurate backward flow. Additionally, there are challenges during the training stage; previous works have often generated suboptimal results by training multi-stage model architectures separately. To address these issues, we propose a Recurrent Flow Update (RFU) model trained in an end-to-end manner. We introduce a global flow update module that leverages global information to mitigate the weaknesses of forward flow and gradually correct errors. We demonstrate the effectiveness of our method through several ablation studies. Our approach achieves state-of-the-art performance not only on the XTest and Davis datasets, which have 4K resolution, but also on the SNU-FILM dataset, which features large motions at low resolution.

## 1. Introduction

Recently, there has been an increasing demand for high-resolution and high-frame-rate videos. As a result, Video frame interpolation technology is being applied in various fields, similar to super-resolution [1,2,3,4,5,6,7,8,9] and denoising [2,10,11,12,13,14], and there is active development of models using deep learning. Video frame interpolation (VFI) is a task that creates an intermediate frame using the two surrounding frames, resulting in slow-motion effects or high-frame-rate videos. For example, information is gathered from both surrounding frames to synthesize the gray frame, and this process continues along the time axis, as shown in Figure 1. VFI has a wide range of applications, from research areas such as video compression, slow-motion generation, and view synthesis to practical devices used in television, streaming, and gaming video. With the increasing prevalence of high-resolution videos, there is a growing need for video frame interpolation (VFI) technologies that can handle 4K resolutions and above. However, research on VFI specifically for high-resolution videos is still limited.

Most pixel-based approaches have utilized Convolutional Neural Networks (CNNs) or transformer architectures [15,16,17,18] to construct various receptive fields, while flow-based approaches have predicted flow maps to warp frames in order to generate an interpolated frame [1,19,20,21,22,23,24,25,26,27,28,29,30,31]. In the early stages of VFI [15,16,17] conducted by CNNs, simple CNN architectures are employed to generate frames directly or to predict weight values for final computations. With the advent of transformer architectures, it became possible to construct a large receptive field through global correlation [18].

In structures that perform VFI via flow estimation, various methods are utilized for flow prediction and refinement. One approach [19,20,21,22,23,26,27] involves constructing a cost volume similar to optical flow estimation algorithms [32,33] to predict flow. In contrast, another approach [24,25,28] conducted by U-Net architectures [34] is used to directly predict flow from input frames, allowing for iterative refinement of the flow. Recently, the adoption of transformer architectures [29,30,31] has led to significant performance improvements by extracting global features, which are then used to predict flow. Nevertheless, these existing methods are not suitable for synthesizing high-resolution videos due to several limitations.

From the perspective of the architectures, when constructing receptive fields for frame generation, the pixel-level synthesis required for 4K video results in high complexity, making it impractical for processing. Additionally, flow-based methods can be categorized into forward warping and backward warping. In the case of forward warping, the generated frames often contain holes. Various approaches try to solve this weakness, but they cannot handle this problem perfectly. On the other hand, backward warping requires prediction of backward flows, which can be obtained from the inputs and intermediate frames. However, since the purpose of VFI is to generate a middle frame, previous approaches cannot produce accurate backward flows.

Another problem is that previous studies have often produced sub-optimal results by training on low-resolution images or not using an end-to-end training approach. In existing approaches, end-to-end training is typically not performed in two cases. The first case [20,21] occurs when multiple modules are connected sequentially and each module is trained separately. For example, in the case of ABME [20], the architecture consists of a motion estimator and a frame synthesis network, and different loss functions are applied to each network, leading to distinct training processes. The second case is when training for high-resolution VFI. There are inherent issues with training solely on low-resolution images. Most studies typically train on the Vimeo90K dataset, which consists of low-resolution images of size 448×256. To address this, several papers [26,29] have proposed training on low-resolution images followed by fine tuning with high-resolution datasets. In the case of SGM [29], for example, the network corresponding to the local branch is first trained on low-resolution images; then, a global branch is added and trained subsequently. However, since these studies did not train their models in an end-to-end manner, they offered sub-optimal solutions for 4K VFI.

To address these problems, we propose two effective approaches. First, we introduce a Recurrent Flow Update (RFU) model with an image pyramid structure, where flow is obtained from lower resolutions and gradually increased to higher resolutions by using a difference map. The difference map indicates which areas are vulnerable in the input image when calculating the flow map. As the resolution increases in accordance with the recurrent structure, this map propagates from low levels to high levels, serving as a guideline for the model. However, as the resolution increases, the area that needs to be processed generally expands. To overcome this limitation, we propose a new sampling method. By utilizing this difference map, we implement a Global Flow Update (GFU) module to enhance the predicted flow with low complexity. This module improves the regions highlighted by the difference map by leveraging global features. Since the sampled difference map varies depending on the video, we propose an efficient method to obtain a map adapted to the characteristics of each video, enabling effective operation, even for 4K high-resolution videos.

Secondly, we employ multiple stages of training, and the proposed model is finally trained in an end-to-end training process. This approach addresses the issue of obtaining suboptimal results mentioned earlier. We apply different training methods at each stage while ensuring that we achieve optimal results by the end. In the first stage, we train a single-level model (detailed in Section 3) using low-resolution images, excluding the GFU module from the RFU model. In subsequent stages, the trained single-level model operates recurrently using the image pyramid, and the GFU module is trained utilizing high-resolution images. Finally, we perform fine tuning through an end-to-end approach on high-resolution images. These approaches lead to remarkable improvements in high-resolution VFI performance. The contribution of this work is summarized as follows:We reveal that existing pixel-based and flow-based methods are not suitable for 4K resolution and propose a new model called the Recurrent Flow Update Model. The proposed method calculates the difference map and effectively samples, which reduces complexity, even in high-resolution images. Additionally, by utilizing a recurrent structure, it propagates together as the level increases, facilitating flow restoration. The global flow update module refines the flow by appropriately retrieving information from global features using the sampled difference map.We conduct training through an end-to-end process to achieve optimal results for 4K VFI. This approach resolves the limitations of existing methods, which either primarily use low-resolution images for training or utilize high-resolution images while training modules separately, leading to suboptimal outcomes.Our proposed algorithm demonstrates effective performance on 4K high-resolution and large motion videos from the X4K1000FPS [35], Xiph [36], SNU-FILM [16], and DAVIS [37] datasets.

## 2. Related Work

Most research on video frame interpolation (VFI) focuses on predicting motion and compensating for the differences between predicting motion and the ground-truth flow based on that prediction. This approach can be broadly categorized into two main strategies: a pixel-based approach that directly generates frames using CNN and transformer architectures and another that initially computes optical flow and refines the predicted flow. On the other hand, unlike typical algorithms, there has been a trend towards combining these methods or conducting VFI for 4K resolutions.

The strategy of directly generating frames or implicitly predicting motion has been widely utilized since the early days of VFI. Niklaus et al. [38] proposed a method that adaptively constructs weights for large kernels for each pixel in the input images. By using these adaptively configured weights, the intermediate frame is generated through convolution operations. However, due to the high computational load, Niklaus et al. [39] effectively reduced the computational burden by employing separable kernels. Despite this, kernel-based methods face limitations because they create a receptive field proportional to the kernel size, making efficient computation challenging. To address this issue and enable more efficient operations, Lee et al. [15] introduce a novel approach using deformable convolution. In contrast, Choi et al. [16] utilized attention modules to directly generate frames, resulting in sharper images. After the introduction of transformer architectures, Lu et al. [30] pointed out the limitation of convolution operations in CNNs, which can only capture a narrow range of motion, and proposed a solution using transformers. Shi et al. [18] expanded upon the structure proposed by Lee et al. [15] by incorporating transformer components into their model. However, these approaches still struggle with the inherent issue of increased computational demands, making them generally impractical for high-resolution applications.

Algorithms for video frame interpolation (VFI) that predict optical flow and use it for compensation have evolved in various ways. Among these, some models generate frames using deep learning techniques to predict optical flow [32,33,40]. Liu et al. [41] and Jiang et al. [42] proposed structures that obtain flow maps by directly predicting the movement of pixels. Unlike the methods proposed in [41,42], which employ their own flow map, structures utilizing pre-trained optical flow models have been proposed. However, there are inherent issues with using these networks. Since the intermediate frame does not exist, the flow must be derived from the input, which necessitates the use of forward warping. If forward warping is applied naively, it can result in holes in the output image, leading to the development of subsequent methods aimed at addressing this issue. Bao et al. [43] introduced a depth estimation module to mitigate the drawbacks of forward warping, thereby improving the quality of the generated results. Niklaus et al. [28] employed a softmax function to facilitate pixel warping at the same location, a technique that has been widely adopted in later studies. Park et al. [21] initially assumed symmetric motion during prediction, which yielded good performance. However, Park et al. [20] later identified the limitations of symmetric motion prediction and proposed a new structure that robustly operates in VFI through asymmetric motion prediction. Kong et al. [24] integrated warping and refinement into a single network architecture, demonstrating its efficiency compared to previous approaches. Huang et al. [25] avoided forward warping by eliminating the pre-trained optical flow network and, instead, proposed a model that predicts flow starting directly from the intermediate frame, achieving near real-time performance. Li et al. [27] and Jin et al. [19] utilized a cost volume structure similar to that used in optical flow models, gradually scaling up to refine the flow map. Subsequently, methods that integrate flow-based approaches have been introduced, with Zhang et al. [31] leveraging the ability of these structures to extract global features for direct flow estimation. Despite these advancements, the previously proposed methods have not completely overcome the issues associated with generating incomplete flows and have not adequately addressed flow prediction at high resolutions, making them unsuitable for high-resolution VFI applications.

On the other hand, recent research has focused on how to effectively enhance optical flow for high-resolution images. Sim et al. [35] adopted a method that increases resolution by repeatedly passing through multiple hierarchical structures. As the resolution increases, the flow is refined multiple times, and a lightweight model is constructed through parameter sharing among different components. Park et al. [26] first predicted global motion, then symmetrically refined the flow, training each proposed upsampling module separately to achieve remarkable performance on high-resolution images. Liu et al. [29] trained a local branch on low-resolution data and added a global branch capable of predicting high-resolution flow to synthesize images. However, a limitation of these previous studies is that each module was trained independently, resulting in suboptimal outcomes.

## 3. Method

### 3.1. Overall Process

The goal of video frame interpolation (VFI) is to generate an intermediate frame ($I_{t}$) at a specific time step (*t*) using the two given consecutive frames ($I_{0}$ and $I_{1}$). The value of *t* has a range of [0,1], and we aim to generate the midpoint frame, which corresponds to t=0.5. Specifically, our model is designed to perform well on high-resolution videos with large motion, such as 4K videos. The overall structure of our network is illustrated in Figure 2. The number of times the proposed network runs recurrently is indicated by the level (l=0,1,2…), and the value of the level changes according to the size of the resolution received as input. The determination of the level value is given by the following equation:(1)$$l=ceil(min\left(log_{2}\left(H\right),log_{2}\left(W\right)\right))-7,$$
where ceil() rounds up a float number to an integer. Therefore, as the resolution of the input increases, the value of the level also increases. In Figure 2, for the sake of convenience in explanation, the level value is set to 3, and the size of the resolution corresponding to level 0 in the image pyramid is 1/4. To optimize this network for high-resolution VFI, we conduct end-to-end training through three steps.

Step 1: In this step, we train only the single-level module without the global flow update (GFU) module and global feature extractor shown in Figure 2. In step 1, since training is conducted only for a single level, *l* is set to 0. First, the Feature Extractor (FE) module takes the input frames ($I_{0}^{l}$ and $I_{1}^{l}$) to predict features ($f_{0,1}^{l}$) as shown in the following equation:(2)$$f_{0,1}^{l}=FE\left(I_{0}^{l},I_{1}^{l}\right).$$

Second, the IF block predicts the backward flow (${\hat{F}}_{t\to 0}^{l},{\hat{F}}_{t\to 1}^{l}$) and the occlusion mask ($M^{l}$). In the IF block, we downsample the input frames and feed them into convolutional neural layers with $f_{0,1}^{l}$ to predict the optical flow and the occlusion mask. Thereafter, we upsample the predicted flow and mask to return them to their original scale in order to synthesize the intermediate frame using the following equation:(3)$$\begin{array}{cc} \; & I_{W,0}^{l}=W\left(I_{0}^{l},UP\left({\hat{F}}_{t\to 0}^{l}\right)\right)\\ \; & I_{W,1}^{l}=W\left(I_{1}^{l},UP\left({\hat{F}}_{t\to 1}^{l}\right)\right)\\ \; & I_{S}^{l}=UP\left(M^{l}\right)\cdot I_{W,0}^{l}+\left(1-UP\left(M^{l}\right)\right)\cdot I_{W,1}^{l} \end{array}$$
where W and UP indicate the backward operation and the bilinear upsampling operation, respectively.

Finally, we synthesize the intermediate frame ($I_{t}$) using the Synthesis Net module. We extract features from the original frames ($I_{0}^{l},I_{1}^{l}$) from the image pyramid and warp them with the refined backward flow (${\hat{F}}_{t\to 0,1}^{l}$). Then, we use a simple U-Net structure to combine features from each module and predict the interpolated frame ($I_{t}$).

Step 2: In this step, we introduce additional modules to the single level trained in step 1 to enhance its robustness against large motions in high-resolution videos. Additionally, because high-resolution images are used for training, the number of levels increases. We add the global flow update (GFU) module and global feature extractor, which are indicated in purple in Figure 2.

First, we update the backward flows (${\hat{F}}_{t\to 0}^{l}$, ${\hat{F}}_{t\to 1}^{l}$) predicted by the IF block in step 1 using the global flow update module. To do this, the global feature extractor takes the input frames ($I_{0}^{l}$ and $I_{1}^{l}$) and predicts the global features ($g_{0}^{l}$ and $g_{1}^{l}$), which consider global correlation. The backward flows (${\hat{F}}_{t\to 0}^{l}$, ${\hat{F}}_{t\to 1}^{l}$) are inaccurate because they estimate an intermediate frame that does not exist. Therefore, a difference map ($D^{l}$) is generated to identify the areas that are lacking. This difference map is computed at each level and, like the flow, continues to be used as the level increases. The difference map is designed to identify damaged areas, so its values and area decrease as the level increases. However, directly using the difference map can become complex due to the increased resolution. To address this, we employ a new sampling method that focuses on processing only the important parts of the difference map. The process of sampling the difference map is described in detail in Section 3.3. After obtaining the difference map, global matching is performed using the difference map and the global features ($g_{0,1}^{l}$) obtained from the global feature extractor to enhance the predicted flows (${\hat{F}}_{t\to 0,1}^{l}$). We utilize the obtained difference map ($D^{l}$) in global matching of the GFU module and in the FE module at the next level. The process of updating the flow is described in detail in Section 3.2. After processing the GFU module, we obtain the interpolated frame ($I_{t}$) using the same Synthesis Net module as in step 1. In step 2, we freeze the learnable parameters of the FE module and the IF block, training only the additional modules, along with the Synthesis Net module. Before passing through the Synthesis Net at each level (*l*), the results (${\hat{F}}_{t\to 0,1}^{\prime l},M^{\prime l},D^{l}$) are used as inputs for the next level (l+1), allowing for recurrent training. All blocks at each level share the same weights.

Step 3: In steps 1 and 2, each module is trained separately. As a result, the modules trained in step 1 operate without incorporating the functions of the modules trained in step 2, leading the overall model to predict suboptimal interpolated results. To address this issue, we release all learnable parameters and fine tune the entire model at a lower learning rate. This approach enables the overall model to perform frame interpolation optimally.

### 3.2. Global Flow Update Module

In this subsection, we describe the global flow update (GFU) module. As shown in Figure 3, the purpose of the GFU module is to transform the predicted flow (${\hat{F}}_{t\to 0,1}^{l}$) into a refined flow (${\hat{F}}_{t\to 0,1}^{\prime l}$) using global information. First, we perform sampling. This process identifies which parts of the flow are inaccurate for each image, and the results are represented as a difference map ($D^{l}$). The difference can be obtained by shifting the predicted flow (${\hat{F}}_{t\to 0,1}^{l}$) of each image, then returning to the original position to calculate the difference. The method for obtaining the difference map is expressed as follows:
(4)$${\hat{I}}_{1}^{l}=W_{F}\left(W_{B}\left(I_{0}^{l},{\hat{F}}_{t\to 0}^{l}\right),{\hat{F}}_{t\to 1}^{l}\right),\hat{J}=W_{F}\left(W_{B}\left(J,{\hat{F}}_{t\to 0}^{l}\right),{\hat{F}}_{t\to 1}^{l}\right),$$
(5)$${\hat{D}}_{0\to 1}^{l}=\sum \left|I_{1}^{l}-{\hat{I}}_{1}^{l}\right|,M_{0\to 1}^{hole}=\sum \left|J-\hat{J}\right|,$$
(6)$$D_{0\to 1}^{l}=M_{0\to 1}^{hole}⊙{\hat{D}}_{0\to 1}^{l},D_{0}^{l}=W_{F}\left(W_{B}\left(D_{0\to 1}^{l},{\hat{F}}_{t\to 1}^{l}\right),{\hat{F}}_{t\to 0}^{l}\right),$$
where $W_{F}$ and $W_{B}$ indicate forward and backward warping, respectively, and ⊙ and *J* are element-wise multiplication and one matrix. By realigning the image ($I_{0,1}^{l}$) using the predicted flow (${\hat{F}}_{t\to 0,1}^{l}$), we can obtain $D^{l}\left(D_{0}^{l},D_{1}^{l}\right)$. However, as the level increases, the resolution that needs to be processed also generally increases. Since the obtained $D^{l}$ indicates which areas are lacking in the image, the number of pixels increases exponentially with each level. To address this issue, we propose a new sampling method. Instead of specifying a percentage of the total number of pixels for sampling, we determine the number of sampling pixels based on the distribution of each difference map. This approach allows for high-resolution VFI tailored to the unique characteristics of each image while reducing the computational burden that arises with increasing resolution. The detailed process is introduced in the Section 3.3.

Next, we perform global matching using the $D^{l}$ obtained from the sampling process and the global feature ($g_{0,1}^{l}$) extracted from the global feature extractor. During global matching, we directly predict the flow indicated by $D^{l}$ using attention. At this stage, we extract only the regions corresponding to $D^{l}$ from the global feature to obtain the flow for the damaged areas. A brief equation representing this process is provided below:(7)$$f_{0\to 1}^{l},f_{1\to 0}^{l}=Attn\left(topk\left(g_{0,1}^{l},D^{l}\right),g_{0,1}^{l}\right),$$
(8)$$F_{t\to 1}^{l,match}=W_{F}\left(\left(1-t\right)f_{0\to 1}^{l},tf_{0\to 1}^{l}\right),F_{t\to 0}^{l,match}=W_{F}\left(tf_{1\to 0}^{l},\left(1-t\right)f_{1\to 0}^{l}\right),$$
where Attn refers to the scaled dot-product attention mechanism and topk indicates the selection of the top k points guided by $D^{l}$. Here, the method for determining *k* in the topk is based on the proposed sampling method. In global matching, the predicted flow is matched against the damaged areas using $D^{l}$ as a guideline. As a result, the output ($F_{t\to 0,1}^{l,match}$) can directly compensate for the damaged regions of the predicted flow. Similar to sampling, since the output is generated through forward warping, the final update is performed using the subsequent merge and refine modules.

Finally, we combine $F_{t\to 0,1}^{l,match}$ and ${\hat{F}}_{t\to 0,1}^{l}$ in the merge module to fill in the damaged areas of the predicted flow, guided by the difference map ($D^{l}$). Then, we refine the merged flows using the refine module, which consists of convolutional neural layers. We warp the input frames ($I_{0,1}^{l}$) using the merged flows ($F_{t\to 0,1}^{l,merge}$). The refine module takes the warped frames, merged flows, predicted mask ($M^{l}$), and predicted flows (${\hat{F}}_{t\to 0,1}^{l}$) to predict the residual, $\Delta F_{t\to 0,1}^{\prime l}$, and $\Delta M^{\prime l}$, which are utilized to obtain the refined flows and mask with the following equation:(9)$${\hat{F}}_{t\to 0,1}^{\prime l}={\hat{F}}_{t\to 0,1}^{l}+\Delta F_{t\to 0,1}^{\prime l},$$
(10)$$M^{\prime l}=M^{l}+\Delta M^{\prime l}$$

The results obtained through this process are ${\hat{F}}_{t\to 0,1}^{\prime l}$, $D^{l}$, and $M^{l}$, which are used as input in the next level.

### 3.3. Sampling Difference Map

This subsection introduces an effective method for obtaining the sampling difference map ($D^{l}$), as explained in Section 3.2. $D^{l}$ represents the incomplete areas when predicting flow in images, and a higher value of $D^{l}$ indicates a greater error. Therefore, the $D^{l}$ values are not uniform across all images, and the distribution of $D^{l}$ varies for each case. To address this, we sample $D^{l}$ differently based on its distribution and adjust the sampling method according to the level. This approach helps manage the increase in computational cost as the level rises.

Figure 4 illustrates the distribution of $D^{l}$ for two arbitrary images, (a) and (b). This distribution is sorted by the value of the difference map, then plotted as a graph. The vertical axis represents the values of $D^{l}$, while the horizontal axis corresponds to the index of pixels based on the resolution of $D^{l}$. As can be seen in the figure, the distributions of the two images differ from each other. This indicates that when sampling based on the number of pixels, such as sampling 50% of the resolution, the distribution of $D^{l}$ is not considered. Additionally, when sampling is conducted as a percentage of the total number of pixels, the complexity increases exponentially as the resolution increases. Therefore, to adopt an approach tailored to each image and reduce complexity as the image resolution increases, we perform sampling based on the integral values derived from the distribution of $D^{l}$ obtained from each image. For example, even when sampling with the same area ratio, image (a) has values concentrated in a narrower range compared to image (b), allowing for sampling with fewer pixels. This method is more effective than simply extracting by cutting based on the number of pixels, and it provides an effective guideline in attention modules, where complexity increases with higher resolution.

The sampling method at each level is as described above, and the reference value for sampling based on width varies depending on the level. The sampling rate at each level is defined as $80/2^{l}\left(\%\right)$. According to this formula, as the resolution increases, the sampling rate decreases, allowing the global flow update module and feature extractor to operate with lower computational loads.

## 4. Experiments

### 4.1. Implementation Details

We train our model in three stages. In the first and second stages, we train each specific module separately, and in the final stage, we train the entire model in an end-to-end manner.

Stage 1: In the first stage, we train a single-level block without the global flow update (GFU) module and global feature extractor. To train this network, we utilize the Vimeo90K [44] dataset, which contains 51,312 sequences with a resolution of 448×256. We perform data augmentation on the training dataset by randomly cropping it to a size of 256×256, followed by random flipping, time reversal, and rotation. We use a batch size of 32 and update the learnable parameters using the AdamW optimizer. Additionally, we gradually increase the learning rate to 2 $\times \;10^{-4}$ over 2000 iterations, then train the network for 300 epochs while decreasing the learning rate from 2 $\times \;10^{-4}$ to 2 $\times \;10^{-5}$ using cosine annealing. To optimize the network, the following loss function is utilized.
(11)$$L=L_{lap}\left(I,I_{pred}\right)+0.5\cdot \sum_{i}L_{lap}\left(I,I_{S}^{i}\right),$$
where $I_{pred}$ is an output of the proposed network, $L_{lap}$ indicates Laplacian pyramid loss [28], and *i* represents the index of the synthesized image ($I_{S}^{i}$).

Stage 2: In this stage, we train the GFU module, global feature extractor, and Synthesis Net while freezing the remaining learnable parameters of the network trained in stage 1. To train these modules for high-resolution videos, we use the X4K1000FPS (X-Train) [35] dataset, which has a higher resolution. This dataset consists of 4408. sequences at a resolution of 768 × 768, with each sequence comprising 65 frames. We perform data augmentation on the dataset by cropping it into 512 × 512 patches, then randomly flipping them. We gradually increase the learning rate to 2 $\times \;10^{-4}$ over 1000 iterations, then decrease it to 2 $\times \;10^{-5}$ using cosine annealing over 100 epochs. Moreover, we use the same loss function as in stage 1.

Stage 3: In this stage, we fine tune the entire network in an end-to-end manner. For the training of this network, we use the same dataset and pre-processing as in stage 2. The learning rate is gradually increased to 1 $\times \;10^{-5}$ over 1000 iterations, then decreased to 1 $\times \;10^{-6}$ over 100 epochs using cosine annealing. The loss function is defined as follows:(12)$$L=L_{lap}\left(I,I_{pred}\right).$$

### 4.2. Test Dataset

We select four test benchmarks that contain videos with large motion or high resolution. We then evaluate each algorithm by comparing the PSNR (Peak Signal-to-Noise Ratio) and SSIM (Structural Similarity) [45] for selected datasets. The details of each dataset are outlined as follows.

DAVIS 2017 [37] is a frequently used dataset for video object segmentation, containing a variety of images ranging from 1280 × 720 to 4K resolution. This dataset has high-quality frames for various situation. It contains 30 sequences, of which only the 4K dataset is used for evaluation.X4K1000FPS (X-Test) consists of 4K resolution data and is composed of 15 scenes. Each scene is determined based on the magnitude of optical flow from entire the X4K1000FPS dataset, and each is grouped into sets of 32 frames. The first and last frames of each scene are used for evaluation.Xiph [36] consists of eight scenes, each containing approximately 100 frames. A new test set is constructed by sampling from the original Xiph dataset, following the methodology used in SGM-VFI [29]. Since the Xiph videos are extracted at 60 fps, which does not match the fps of X-Test, the frame numbers are doubled for evaluation to ensure consistency. The sampled pairs used for evaluation are referred to as Xiph-L and consist of 192 frames.SNU-FILM [16] is a dataset categorized into four difficulty levels (easy, medium, hard, and extreme) based on the temporal distance between input frames. This dataset consists of the test split of the GOPRO dataset using realistic image pairs using a high-speed camera sensors and manually collected video sequences from YouTube. The dataset has a resolution of 1280 × 720 and contains 310 pairs of images, each consisting of three images. For evaluation, only the hard and extreme sets, which represent higher difficulty levels, are used. The hard and extreme datasets are denoted as SNU-FILM-H and SNU-FILM-E, respectively.

### 4.3. Quantitative Results

We compare the proposed method with state-of-the-art algorithms such as XVFI [35], RIFE [25], IFRNet [24], VFIformer [30], AMT [27], UPR [19], EMA [31], Biformer [26], and SGM-VFI [29]. Table 1 shows the results of comparisons between previous methods and our proposed network. *OOM* indicates out of memory, and the units for PSNR and runtime are dB and seconds, respectively. In the case of runtime, we evaluate how much time is required to interpolate the 4k frames on an NVIDIA RTX A6000. As shown in Table 1, we achieve state-of-the-art results on almost all test datasets, and also we maintain a lightweight model compared with previous studies. Especially on the test datasets, where performance is higher compared to other algorithms, there is an improvement of about 0.2 dB across the board. This indicates that the proposed model operates effectively in 4K resolution VFI compared to existing algorithms. Furthermore, even on the SNU-FILM dataset, which has a low resolution but contains large or complex motion, we observe that our proposed model demonstrates top performance. This confirms that our model is robust not only for high resolutions but also for large motion in various sizes of videos.

### 4.4. Qualitative Results

Figure 5 visualizes the results of performing VFI at 4K resolution. The visualized images are sourced from the DAVIS, Xiph, and X4K1000FPS datasets. The first column represents the ground-truth frames that need to be generated, and the sections highlighted by red boxes are enlarged and displayed on the right. Overall, the results indicate that previous studies aimed at 4K fail to maintain the structure of objects in several cases. In particular, existing methods fail to generate the wrist of the human in the third case and the grid pattern of the iron bars or the mark of the truck in the last case. This indicates that relatively small objects and texture information are not effectively reproduced in high-resolution scenarios. In contrast, the recurrent flow update (RFU) network demonstrates significantly better results compared to previous algorithms.

Figure 6 presents results for low-resolution videos with large motion. Unlike Figure 5, in which a model designed for 4K resolution is used, this comparison includes algorithms that are not specifically focused on 4K [24,25]. The large image on the left overlays the two input images to illustrate the extent of motion, and the regions of overlapped inputs highlighted by red boxes are displayed on the right. In the first image, while existing models show a significant distortion of the structure of the car, our proposed model presents results that closely resemble the ground truth. In the second video, we can observe large motion as a bird flaps its wings. Other models encounter issues such as missing or duplicated wings, whereas our proposed model accurately represents the wings as a single object. This demonstrates that our approach is not only effective at 4K resolution but also robust in handling videos with large motion.

## 5. Discussion

This section discusses three key elements of the proposed method. Each subsection analyzes how the respective technique has influenced the results.

### 5.1. Multi-Level Architecture

As mentioned in Section 3, the proposed recurrent flow update (RFU) network has a structure that incrementally refines the flow through multiple levels. To determine the effectiveness of this structure, we train a new single-level structure and compare the results. In Table 2, ‘w/o multi-level’ refers to the results obtained by training the single-level structure. Overall, the performance is lower compared to that of the final model, demonstrating that the single-level structure struggles to handle high-resolution images and large motions. Therefore, it can be concluded that the proposed recurrent updating flow architecture operates effectively.

### 5.2. End-to-End Training

To address the issue of obtaining suboptimal results, which has been pointed out as a problem in previous methods, we propose a training strategy consisting of three steps. To analyze the impact of this approach, we compare the results with a structure trained only up to step 2. As shown in Table 2, the performance of the structure trained without end-to-end training is approximately 0.2 dB lower in terms of PSNR compared to the structure trained through all three steps.This demonstrates that the VFI results obtained without end-to-end training are suboptimal.

### 5.3. Global Flow Update Module

The global flow update (GFU) module is a crucial component that compensates for the degradation of predicted flow (${\hat{F}}_{t\to 0,1}^{l}$) by acquiring global information from the global feature extractor. To assess the impact of this module, we compare the performance of the model trained without the GFU. As seen in the last row of Table 2, the overall performance is lower in terms of PSNR. Nevertheless, since quantitative evaluation does not effectively reveal the impact on the generation of high-resolution images, we also conduct a qualitative assessment. Figure 7 visualizes the results by selecting one image from Figure 6. The model without a GFU module struggles to capture global information, which makes it challenging to restore vehicles with large motion. Overall, this indicates that the GFU module plays a vital role in enhancing flow to ensure robustness against high-resolution images and large motion.

### 5.4. Sampling Difference Map

As mentioned in Section 3.3, when sampling the difference map ($D^{l}$), we set different rates depending on the level. To evaluate this effect, we extract and compare the difference maps from the trained model according to the levels. Figure 8 shows the magnified difference maps at each level. When comparing levels 0 and 1, there is a tendency for the area indicated in the difference map to decrease and become more detailed as the level increases. This is because, in the process of progressively refining the flow, the range of damage decreases as the level rises. At the same time, the sampling rate is learned to decrease, leading to a concentration of the difference map distribution around specific pixels. This indicates that the proposed method reduces the computational load required for higher levels, which correspond to higher resolutions. Therefore, we can confirm that the proposed sampling method is effective for high-resolution VFI.

To verify the efficiency of the proposed sampling method, a comparison is made with a pixel-based sampling approach. Table 3 presents the runtime and performance results. ‘Pixel sampling’ refers to the sampling conducted based on the number of pixels, applying the sampling rates ($80/2^{l}\left(\%\right)$) according to the levels (*l*) presented in Section 3.3. In contrast, ‘50% pixel sampling’ samples 50% of the pixels for each resolution, regardless of the level. As shown in the results reported in Table 3, the overall performance is similar. In terms of runtime, despite the similar performance, the proposed sampling method is the fastest. This indicates that sampling based on the integral values derived from the distribution of the difference map efficiently reflects the importance of the map.

## 6. Conclusions

Existing video frame interpolation algorithms have significant limitations in handling high-resolution videos due to incomplete flow maps and computational complexity. To address this, we propose a recurrent flow update (RFU) model that enhances the flow through multiple stages. Among these, we introduce the global flow update (GFU) module, which incorporates global information for improvement, and we present a new sampling method for the difference map used in this module, allowing it to operate efficiently, even at high resolutions, without computational issues. Finally, we train the entire model in an end-to-end manner to fully optimize the network. When evaluated on various high-resolution datasets, our proposed model demonstrate strong performance both quantitatively and qualitatively, proving that the RFU network is suitable for 4K video frame interpolation.

## Figures and Tables

**Figure 1 sensors-25-00290-f001:**
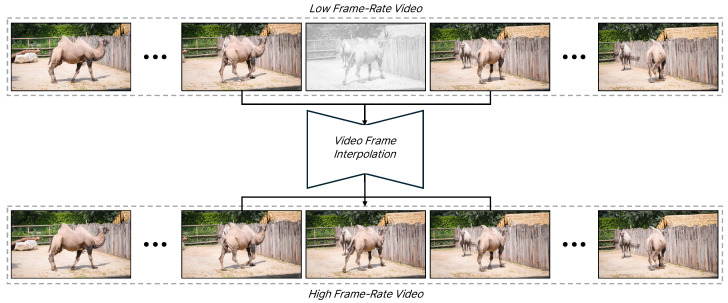
The process of video frame interpolation. This figure illustrates the application of video frame interpolation to low-frame-rate videos. The gray image represent frames that do not actually exist; these frames are generated by referencing the adjacent frames through the video frame interpolation process. By repeating this along the time axis, a high-frame-rate video can be obtained.

**Figure 2 sensors-25-00290-f002:**
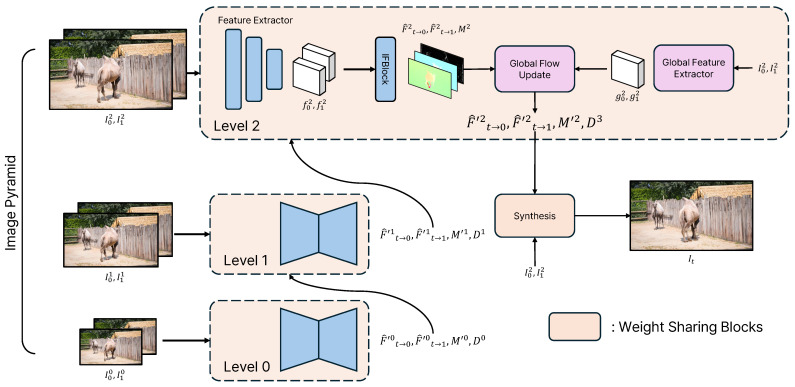
Overall structure of the recurrent flow update (RFU) network. The blocks at each level marked in orange have the same structure. In the first training step, we train a single-level block without the global flow update (GFU) module and global feature extractor. Thereafter, we add and train only the red modules. In the final training step, we train all modules in an end-to-end manner.

**Figure 3 sensors-25-00290-f003:**
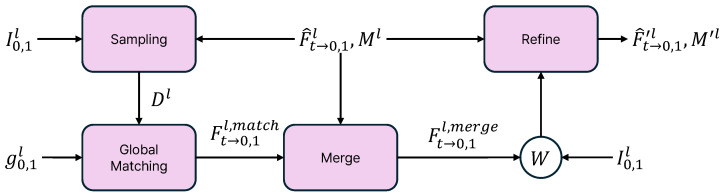
The overall structure of the global flow update (GFU) module.

**Figure 4 sensors-25-00290-f004:**
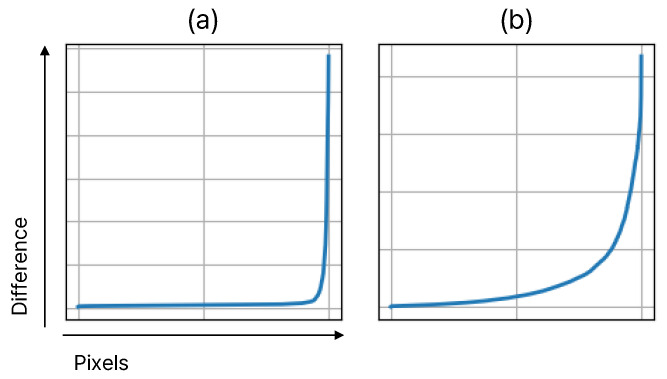
The distributions of $D^{l}$ of two different images. The subfigures (**a**,**b**) represent the distribution of difference maps from two different images. “Pixel” refers to the number of pixels in the image, while “difference” indicates the difference value at each pixel.

**Figure 5 sensors-25-00290-f005:**
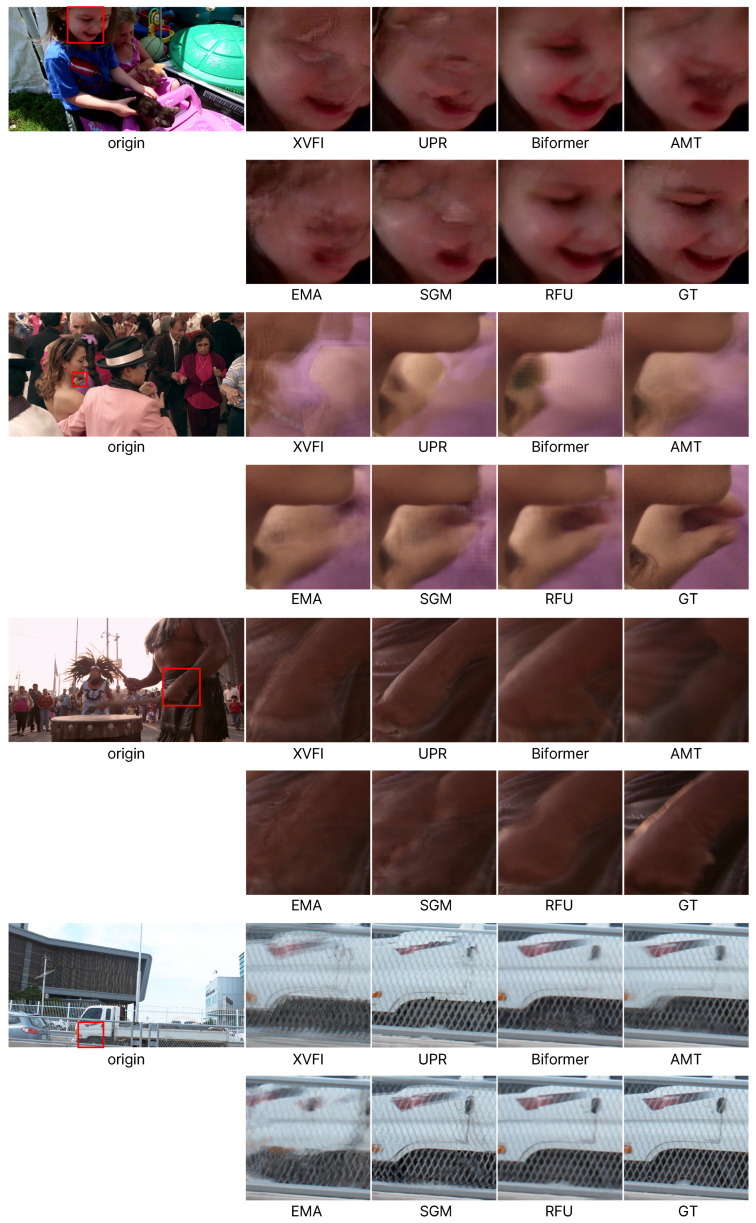
Qualitative results on 4K datasets (DAVIS, X4K1000FPS, and Xiph).

**Figure 6 sensors-25-00290-f006:**
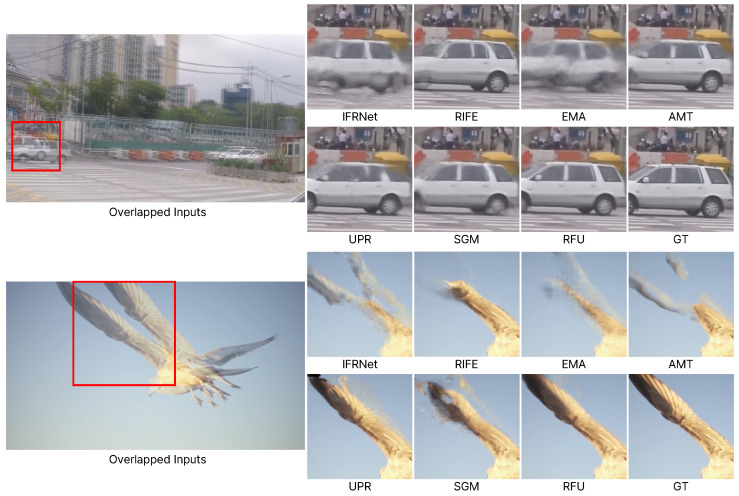
Qualitative results on the SNU-FILM-Extreme dataset.

**Figure 7 sensors-25-00290-f007:**
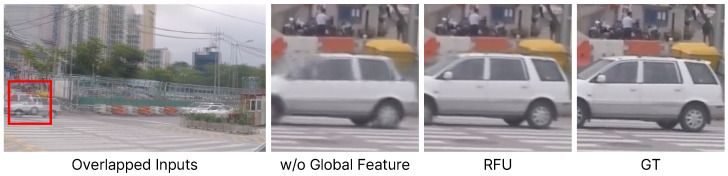
Comparison between ablation studies.

**Figure 8 sensors-25-00290-f008:**
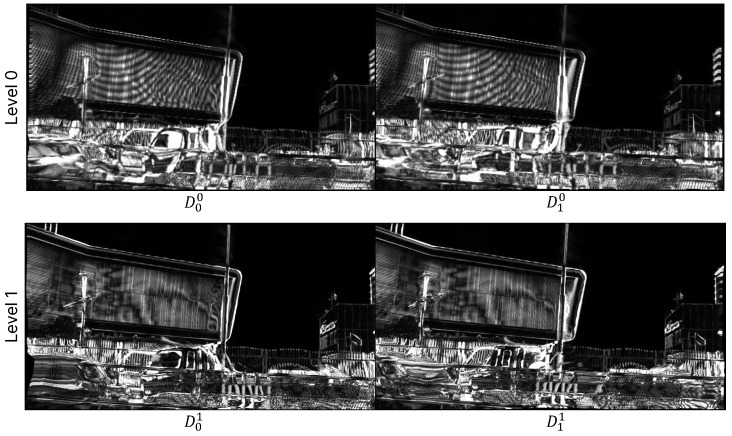
Comparison of multi-level difference maps. The areas marked in white represent regions with high values, while the black areas are close to zero.

**Table 1 sensors-25-00290-t001:** Quantitative evaluation on four datasets. Each value indicates PSNR(dB)/SSIM, respectively. The red highlights define the best results, and blue highlights indicate the second best results.

	DAVIS	X4K1000FPS	SNU-FLIM-H	SNU-FLIM-E	Xiph	Runtime
XVFI [35]	23.12/0.829	25.36/0.836	29.15/0.926	24.55/0.848	26.65/0.886	17.7 s
RIFE [25]	24.07/0.855	25.36/0.801	29.95/0.931	24.89/0.856	27.27/0.887	16.9 s
IFRNet [24]	23.07/0.841	23.04/0.778	29.89/0.932	24.58/0.851	24.17/0.853	8.0 s
VFIformer [30]	*OOM*	*OOM*	30.18/0.934	24.80/0.857	*OOM*	-
AMT [27]	25.67/0.876	30.12/0.900	30.38/0.936	25.05/0.859	30.55/0.918	16.3 s
UPR [19]	25.08/0.864	28.78/0.878	30.64/0.936	25.46/0.863	29.37/0.907	16.6 s
EMA [31]	25.25/0.872	28.88/0.880	30.38/0.934	25.15/0.858	29.87/0.916	16.7 s
Biformer [26]	25.47/0.875	30.11/0.907	-	-	29.35/0.916	18.1s
SGM-VFI [29]	25.43/0.874	29.91/0.897	30.67/0.937	25.51/0.855	29.97/0.915	16.4 s
Ours	25.80/0.879	30.36/0.901	30.84/0.938	25.77/0.869	30.51/0.921	16.2 s

**Table 2 sensors-25-00290-t002:** The evaluation results of ablation studies.

	DAVIS	X4K1000FPS	SNU-FLIM-H	SNU-FLIM-E
w/o multi-level	25.45/0.876	29.47/0.888	30.49/0.934	25.44/0.863
w/o End-to-End Training	25.60/0.877	30.15/0.901	30.65/0.936	25.56/0.864
w/o GFU module	25.76/0.875	29.98/0.897	30.71/0.939	25.67/0.869
Ours	25.80/0.879	30.36/0.901	30.84/0.938	25.77/0.869

**Table 3 sensors-25-00290-t003:** The evaluation results of ablation studies for sampling methods.

	DAVIS	SNU-FLIM-H	SNU-FLIM-E	Xiph	Runtime
50% pixel sampling	25.76/0.878	30.84/0.938	25.77/0.867	30.53/0.921	16.9 s
pixel sampling	25.77/0.878	30.84/0.938	25.77/0.869	30.54/0.921	17.0 s
Ours	25.80/0.879	30.84/0.938	25.77/0.869	30.51/0.921	16.2 s

## Data Availability

Publicly available datasets were analyzed in this study [16,35,36,37].

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
