# Peer review of "Recurrent Flow Update Model Using Image Pyramid Structure for 4K Video Frame Interpolation"

_sensors, 2025, doi:10.3390/s25010290_

Round 1

Reviewer 1 Report

Comments and Suggestions for Authors

The authors proposed a novel Recurrent Flow Update (RFU) model for video frame interpolation (VFI) to address limitations in high-resolution video synthesis. They introduced a Global Flow Update module to leverage global information, enhancing the accuracy of frame interpolation and error correction. The RFU model is trained end-to-end, aiming to improve upon the suboptimal results of previous multi-stage training approaches.

Concerns:

1. One of the main innovations of the paper is the end-to-end approach, but I am curious if training in multiple steps can truly be considered end-to-end?

2. Your method claims to be efficient, and it would be helpful to see model metrics like parameters and runtime in Table 1, rather than just performance metrics.

3. The sampling method you designed is claimed to be computationally efficient, but it seems that there is no experimental validation for this.

4. Figure 3 doesn’t seem to make it easier to understand your method; it’s difficult to establish connections between the numerous symbols and the diagram.

5. I’m curious about the basis for the numbers in Equation 1 and line 295.

Comments on the Quality of English Language

The English could be improved to more clearly express the research.

Reviewer 2 Report

Comments and Suggestions for Authors

Summary:

The paper introduces a novel hierarchical RFU model for high-resolution video frame interpolation, addressing key challenges faced by traditional flow-based methods. The innovative design, complemented by a robust training strategy, leads to significant advancements in interpolation accuracy and resolution handling. While the model demonstrates impressive results on benchmark datasets, its generalizability to real-world scenarios and computational efficiency require further exploration. Despite these limitations, the paper represents a substantial contribution to the field, offering valuable insights and a strong foundation for future research in high-resolution video processing.

Strengths:

1. nnovative Design: The RFU model combines hierarchical flow refinement and the GFU module to address limitations in high-resolution video interpolation, achieving notable improvements over prior methods.

2. Effective Validation: Comprehensive experiments on benchmark datasets demonstrate state-of-the-art performance, with clear quantitative and qualitative results highlighting its robustness in handling large motion and high resolution.

3. Strong Design and Training Strategy: The three-stage training strategy, consisting of single-level training, high-resolution module training, and final end-to-end fine-tuning, ensures the network’s optimal performance. This approach effectively resolves the suboptimal results associated with previous training practices in similar works.

Weaknesses:

1.   Limited Real-World Testing: The model lacks validation on real-world scenarios, such as noisy or dynamically lit videos, which raises concerns about its robustness beyond curated datasets.

2. Computational Complexity: The method’s high computational cost for 4K video interpolation may hinder its deployment in resource-constrained environments, but no efficiency comparison with lighter approaches is provided.

3. Insufficient Comparison with Modern Approaches: Despite referencing transformer-based methods, the paper does not comprehensively compare their performance or computational trade-offs with the proposed method. Such a comparison would help to better contextualize the novelty and strengths of the RFU model.

4. Some important references [1-3] are missing. Please discuss them in the related works.

[1] Zhang, Guozhen, Yuhan Zhu, Haonan Wang, Youxin Chen, Gangshan Wu, and Limin Wang. "Extracting motion and appearance via inter-frame attention for efficient video frame interpolation." In Proceedings of the IEEE/CVF Conference on Computer Vision and Pattern Recognition, pp. 5682-5692. 2023.

[2] Xiao, Jun, Xinyang Jiang, Ningxin Zheng, Huan Yang, Yifan Yang, Yuqing Yang, Dongsheng Li, and Kin-Man Lam. "Online video super-resolution with convolutional kernel bypass grafts." IEEE Transactions on Multimedia 25 (2023): 8972-8987.

[3] Kong, Lingtong, Boyuan Jiang, Donghao Luo, Wenqing Chu, Xiaoming Huang, Ying Tai, Chengjie Wang, and Jie Yang. "Ifrnet: Intermediate feature refine network for efficient frame interpolation." In Proceedings of the IEEE/CVF Conference on Computer Vision and Pattern Recognition, pp. 1969-1978. 2022.

Reviewer 3 Report

Comments and Suggestions for Authors

The paper is well-written and effectively conveys its main arguments, making them easy to understand.

The application of flow-based algorithms to video interpolation is intriguing and demonstrates innovative thinking.

It would be beneficial to refine some of the English expressions to improve the overall readability and clarity of the paper.

Round 2

Reviewer 1 Report

Comments and Suggestions for Authors

Expect to see more about non-end-to-end training methods in the introduction.

Author Response

Comments 1: Expect to see more about non-end-to-end training methods in the introduction.

Response 1: Among existing algorithms, non-end-to-end training is primarily conducted when using various datasets or when multiple modules are connected sequentially. In the introduction section, we introduce the examples for each of these two cases. (line 57-63, line 66-68)

Reviewer 2 Report

Comments and Suggestions for Authors

The authors have addressed my concerns. The quality of the paper is enough for publication.

Author Response

Thank you for the good advice.